# Feasibility of Invasive Brown Seaweed *Rugulopteryx okamurae* as Source of Alginate: Characterization of Products and Evaluation of Derived Gels

**DOI:** 10.3390/polym16050702

**Published:** 2024-03-05

**Authors:** Ismael Santana, Manuel Felix, Carlos Bengoechea

**Affiliations:** Escuela Politécnica Superior, Universidad de Sevilla, Calle Virgen de África, 7, 41011 Sevilla, Spain; mfelix@us.es (M.F.); cbengoechea@us.es (C.B.)

**Keywords:** alginate extraction, gelation, *Rugulopteryx okamurae*, SAOS

## Abstract

*Rugulopteryx okamurae* (RO) is an invasive brown seaweed that causes severe environmental problems in the Mediterranean Sea. This work proposed an extraction method that enables their use as a raw material for producing sodium alginate. Alginate was successfully extracted from this invasive seaweed, with its gelling performance in the presence of Ca^2+^ ions comparable to existing commercial alginates. The mannuronic acid (M)-to-guluronic (G) acid ratio in the ^1^H-NMR profile indicated a higher percentage of G in the RO-extracted alginate, which implies a greater formation of so-called egg box structures. These differences resulted in their different rheological behaviour, as sodium alginate aqueous solutions exhibited a greater viscosity (η at 1 s^−1^ = 3.8 ± 0.052 Pa·s) than commercial alginate (2.8 ± 0.024 Pa·s), which is related to the egg box structure developed. When gelled in the presence of calcium, an increase in the value of the elastic modulus was observed. However, the value of the tan δ for the extracted alginate was lower than that of commercial alginate gels, confirming a structure more densely packed, which implies a different restructuring of the alginate chain when gelling. These results confirm the suitability of using invasive *Rugulopteryx okamurae* as a source of calcium alginate gels. In this way, sustainable bio-based materials may be produced from undesired biomass that currently poses a threat to the ecosystem.

## 1. Introduction

Seaweeds are a natural resource of a wide variety of natural compounds, of which carbohydrates are the most numerous and can reach up to 71.2% of their dry weight, depending on the type of seaweed [1,2]. Most of these carbohydrates are generally found in the form of polysaccharides, whose composition will vary depending on the species of seaweed. Green seaweeds, for example, have a polysaccharide composition similar to that of land plants compared to other algae species [3]. In some species of green seaweeds, cellulose is one of the main polysaccharides that make up their cell wall, apart from other sulphated polysaccharides, such as ulvan or galactan, among others [4]. Regarding red seaweeds, within the wide spectrum of polysaccharides it contains, it is worth highlighting agar, carrageenans and cellulose, among others [5]. These polysaccharides are raw materials usually useful in the industry for film production, where some of the algae used for this purpose are *Eucheuma sp.* or *Palmaria sp.*, among many more [6]. Finally, the polysaccharide composition of brown algae consists mainly of alginate, laminarian, or fucoidan, which have a wide variety of uses such as in the food industry, pharmaceutical, films, etc. Some brown seaweed used for the extraction of these polysaccharides are *Sargassum sp.*, *Laminaria sp.*, or *Undaria sp.* [7,8]. Thus, the extraction of sodium alginate from algae has already been reported; however, the relationship between their structure and functional properties is limited, especially in the case of the invasive species *Rugulopteryx okamurae*.

Alginate is a linear polysaccharide found in the matrix and cell wall of brown seaweeds and is formed by blocks of mannuronic (M) acid and guluronic (G) acids linked by glycosidic bonds. These blocks are found in heteropolymeric sequences along the chain they form, varying their length, order, and position between different alginates [9]. Commercially used alginate is usually found in the form of soluble sodium alginate, for which an extraction process is carried out from the insoluble salts of calcium, potassium, or magnesium alginate that form the cell wall [7,10]. In its soluble salt form, sodium alginate has several uses in the pharmaceutical and food industries thanks to its gelling, coagulant, and encapsulating properties [11]. It is also interesting to highlight the ability of alginate to form gels in the presence of calcium, which has great interest in industry [12].

Although commercial sodium alginate is already available, there are many advantages of using seaweed as a source of this compound. Firstly, they are greatly available and diverse, and they do not compete with other crops. Thus, they do not require land or irrigation for their cultivation, and in many cases, their growth is not uncontrolled. More specifically, the seaweed used in this work was *Rugulopteryx okamurae* (RO). It is an invasive brown seaweed. It is a species of algae that has been considered invasive in Spain since 2020 [13], found mainly in the Mediterranean Sea and western European coasts, currently affecting other countries such as Morocco and France but with prospects of continuing to proliferate [14]. This seaweed, native to the northwest Pacific Ocean, was introduced via the Strait of Gibraltar, where, thanks to the warm waters, a favourable environment is produced for its proliferation [15]. Nowadays, it represents a huge problem since tons of arrivals of RO reach the Mediterranean coasts every year, threatening their use not only for tourism but also the lives of fishers since it pushes away native species, multiplying rapidly and quickly colonizing the Mediterranean coasts [13]. Since the problem cannot be addressed from a biological point of view, finding applications for this seaweed could serve as a path to control their uncontrolled dissemination.

In the present work, the extraction and characterization of sodium alginate from the invasive brown algae RO was carried out. ^1^H-NMR, DSC, TGA, and FTIR analysis were carried out in order to obtain structural information, including its molecular weight, which can be later used to properly justify its properties. Subsequently, aqueous solutions were prepared, and their rheological properties were evaluated and eventually compared with that obtained with commercial sodium alginate by oscillatory rheological tests. Finally, the controlled gelation of calcium alginate was carried out by ion exchange via a dialysis membrane. The mechanical spectra of the calcium alginate gels were obtained. The water absorption and retention properties of freeze-dried gels were assessed by their water uptake capacity and water holding capacity (WUC and WHC, respectively). The results obtained permitted the correlation of the molecular structure of the extracted sodium alginate with the functional properties of the developed gels, in agreement with the egg box structure proposed by other authors.

## 2. Materials and Methods

### 2.1. Alginate Extraction Process

The sodium alginate was extracted according to the scheme represented in Figure 1. This extraction was carried out following the procedure suggested by Calumpong et al. [16] with some modifications. First, 30 mg/mL of milled and freeze-dried seaweed was soaked overnight in 2% formaldehyde to eliminate pigments. The presence of phenolic compounds in the seaweed pigment during the alginate extraction with Na_2_CO_3_ may contribute to the degradation of the polysaccharide. Hence, formaldehyde was employed to generate an insoluble polymer solution, subsequently extracted during the acid treatment [17]. The extract obtained was rinsed with Milli-Q grade water and soaked in a 0.2 M HCl solution for 24 h to protonate the alginic acid. The resulting solid was washed with Milli-Q grade water and stirred for 5 h at 65 °C in a 2% Na_2_CO_3_ solution. The supernatant was collected by centrifugation at 5000× *g* for 30 min, and it was precipitated with ethanol 95% (3 volumes). The solid generated was washed with acetone and dried at 50 °C overnight. Finally, the sample was freeze-dried and stored at −20 °C [16,18,19].

### 2.2. Characterization of Commercial and RO Extracted Alginate

#### 2.2.1. Differential Scanning Calorimetry (DSC)

DSC analysis was carried out by a DSC Q20 (TA Instruments, New Castle, DE, USA) equipped with an autosampler unit and was used to perform DSC. Overall, 5–10 mg samples were weighed; they were placed into hermetic-sealed aluminum pans. Samples were initially equilibrated at −50 °C (10 min), and then, they were heated at 10 °C min^−1^ up to 325 °C with a continuous nitrogen flow (50 mL min^−1^).

#### 2.2.2. Thermogravimetric Analysis (TGA)

TGA analysis was carried out in an SDT Q600 V20.9 Build 20 (TA Instruments, New Castle, DE, USA). The test was performed from 25 to 600 °C at a heating rate of 10 °C/min, with a mass flow of 100 mL/min under N_2_ atmosphere. The weight of the sample was always approximately 10 mg.

#### 2.2.3. Fourier Transform Infrared Spectroscopy (ATR-FTIR)

Extracted and commercial alginate were both measured using an FTIR Invenio X spectrometer (Bruker, Billerica, MA, USA) with an ATR objective. Tests were carried out for values of wavenumber from 4000 to 400 cm^−1^ with a resolution of 4 cm^−1^.

#### 2.2.4. Proton Nuclear Magnetic Resonance (^1^H-NMR)

Solutions containing commercial and extracted alginates (20.2 mg/mL) were prepared in deuterated water (D_2_O 99.9%) and then lyophilized. Subsequently, 100 μL of a solution of TSP-d4 (sodium 2,2,3,3-tetradeutero-3-trimethylsilylpropionate) and 900 μL of D_2_O (99.9%) were added to each lyophilizate, dissolved, and 600 μL was taken for analysis. ^1^H-NMR study was carried out on a Bruker Avance NEO 500 MHz spectrometer (Bruker, Billerica, MA, USA) at 80 °C. A 90° pulse and 32 sweeps were used, with a waiting time between sweeps of 20 s. Overall, 64k points were used to register the free induction decay (FID). ^1^H-NMR spectra were recorded both with and without presaturation of the residual solvent signal (semi-deuterated water, DHO).

#### 2.2.5. Determination of Intrinsic Viscosity and Molecular Mass of the Extracted Alginate

Intrinsic viscosity, [η], is a measure of the hydrodynamic volume occupied by a macromolecule, which depends on the molecular mass and the solvent [20]. Intrinsic viscosity measurements were carried out from sodium alginate solutions in 0.1 M NaCl at concentrations of 0.2, 0.4, 0.6, 0.8, and 1% wt. using a Ubbelhode viscosimeter. NaCl was chosen as the solvent since the constants *k* and *a* used in Equation (2) for the determination of the molecular mass depend on the solvent, and this was the one used by Mark Houwink due to its affinity with the polymer and its reduction in electrostatic repulsions [20,21]. The temperature was kept constant at 25 °C using a Tectron Bio thermostatic bath (J.P. Selecta, Barcelona, Spain). The efflux time of the solvent at the different concentrations prepared was measured in triplicate [21]. Fedors’ Equation (1) was used to relate the concentrations and efflux times with the intrinsic viscosity of the RO-extracted alginate [22]:(1)12(ηr12−1)=1Cη−1Cmaxη
where *η_r_* is the reduced viscosity, which is the ratio of the viscosity of the samples to that of the solvent; *C* is the concentration of each sample; and *C_max_* is a polymer concentration parameter.

Once the intrinsic viscosity of the alginate was determined, the Mark Houwink–Sakurada Equation (2) was used to obtain the molecular mass of the extracted alginate [23].
(2)$$\left[\eta \right]=k{\left(M\right)}^{a}$$
where [*η*] is the intrinsic viscosity, *M* is the molecular mass of the polymer, and *k* and *a* are empirical coefficients dependent on the type of polymer. These coefficients (*k* and *a*, 0.023 and 0.984, respectively) were taken from an empirical regression of several measurements of different alginates from seaweeds (*Fucus vesicularus*, *Laminaria hyperborean* [24], *Laminaria digitata* [25], *Laminaria cloustoni* [26], and *Macrocystis pyrifera* [27]) collected by Clementi et al. (1998) [20].

### 2.3. Gelation Process of Extracted and Commercial Alginate

Alginate gelation was carried out by dialysis using dialysis tubing cellulose membrane 33 mm flat D9652-100FT (Sigma-Aldrich, St. Louis, MO, USA) to obtain a broad and homogeneous structure of approximately 10 cm high and 20 mm in diameter. First, sodium alginate was dissolved in distilled water. Then, this solution was placed in the dialysis tubing cellulose membranes, which were immersed in a CaCl_2_ solution and left to sit overnight. An equivalent amount of glucono-δ-lactone was added to acidify the medium and facilitate the release of Ca^2+^ ions [28,29,30,31].

### 2.4. Characterization of Calcium Alginate Gels

#### 2.4.1. Viscosity of Alginate Solutions

The AR-2000 rheometer (TA Instruments, New Castle, DE, USA) was used to determine the viscosity of the different sodium alginate solutions (1 and 2% wt.) when dissolved in water before gelation. Frequency sweep tests (from 0.1 to 100 Hz) were carried out with low inertia aluminum plates to increase sensibility at 25 °C. All the measurements were carried out within the linear viscoelastic range (LVR), previously determined by strain sweep tests. The temperature was controlled by a Peltier system.

#### 2.4.2. Rheological Properties of Gels

Gelled calcium alginate was cut into smaller pieces 25 mm high and 20 mm in diameter using a scalpel. Frequency sweep tests were carried out with a Haake MARS 40 Rheometer (Thermo Fisher Scientific, Waltham, MA, USA) using parallel rough plates of 20 mm in diameter. Tests were performed from 0.1 to 10 Hz for gels at an alginate concentration of 0.5, 1, and 2% wt. at 20 °C just after obtaining gels. The initial water content of the gels was assessed by measuring the weight difference between the gels obtained immediately after gelation and the lyophilized counterparts, yielding a water content result of 42.16 ± 3.34%.

#### 2.4.3. Water Uptake Capacity (WUC), Soluble Matter Loss (SML) and Water Holding Capacity (WHC)

Starting from freeze-dried gels stored in a desiccator, the first step consisted of drying samples in an oven at 50 °C overnight (20 h) (*w*_1_). Then, they were immersed in distilled water for 24 h and weighed (*w*_2_). Right after immersion, samples were centrifuged at 2000 rpm for 10 min (*w*_3_). Finally, gels were freeze-dried and weighed (*w_4_*). Thus, WUC, SML, and WHC were determined by weight differences following Equations (3)–(5), respectively:(3)$$WUC\;\left(\%\right)=\frac{\left(w_{2}-w_{4}\right)}{w_{4}}\cdot 100$$
(4)$$SML\;\left(\%\right)=\frac{{(w}_{1}-w_{3})}{w_{1}}\cdot 100$$
(5)$$WHC\;\left(\%\right)=\frac{{(w}_{2}-w_{3})}{w_{2}}\cdot 100$$
where *w* indicates the weight, and the subscript number is the step of the process.

#### 2.4.4. Scanning Electron Microscopy (SEM)

Freeze-dried gels were examined by SEM using a ZEISS EVO microscope (ZEISS, Oberkochen, Germany). Samples were coated with an Au-Pd layer. Samples were Au-Pd coated and observed at a working distance of 6 mm, with a bean current of 18 pA and with an acceleration voltage of 10 kV. Images obtained were analyzed by software ImageJ version v1.54i (Bethesda, MD, USA).

### 2.5. Statistical Analysis

Significant differences are indicated with superscript letters (*p* < 0.05). The measurements were carried out at least in triplicate.

## 3. Results and Discussion

### 3.1. Alginate Characterization

The extracted alginate from RO was subjected to different tests in order to (i) ensure its suitability for gelling properties and (ii) compare its performance with respect to commercial samples.

#### 3.1.1. ATR-FTIR and ^1^H-NMR

Figure 2A shows the FTIR profile of the alginate obtained from RO seaweed and commercial alginate. ATR-FTIR spectroscopy can be used to analyze a wide range of organic and inorganic materials, being their FTIR spectrum characteristics since it is relative to their covalent bonding pairs and functional groups ([32]). Alginic acid has a characteristic structure of β-D-mannuronic acid and α-L-guluronic acid that swap the homopolymeric and alternating regions. Although this is the general structure, the concentrations of guluronic and mannuronic acids depend not only on the species but also on ambient conditions such as the season, temperature, and seaweed [33].

Thus, the coincidence in the position of the different peaks observed confirms that the powder obtained following the experimental procedure shown in Figure 1 is sodium alginate. The first peak rose at ~1600 cm^−1^ [34]. This peak, together with the peak observed at ~1400 cm^−1^, corresponds to the vibrations of the O-C-O bond of the carboxylate group, where the former signal can be attributed to asymmetric stretching and the latter symmetric stretching vibration, with a contribution of C-OH deformation vibration [19]. The difference between the heights and the offset may indicate the interference of some impurity from the seaweed. Moreover, both alginates (extracted and commercial) exhibited a peak at ~1026 cm^−1^, which corresponds with the C-O stretching vibrations [35]. It can be mentioned that more intense peaks were obtained for the extracted alginate compared to the ones obtained for commercial alginate at 1600 and 880 cm^−1^. These peaks can be attributed to the presence of sodium carbonate, which has those characteristics peaks too. The small peak found on commercial alginate around 815 cm^−1^ is attributed to mannuronic acid residues [18]. Despite variations in peak heights and phase shifts, the obtained powder is predominantly composed of sodium alginate, as evidenced by the presence of its characteristic peaks. However, as stated above, its performance depends on both structure and ambient condition growth. Further analyses were conducted to assess the performance of the extracted powder.

Figure 2B shows the ^1^H-NMR analysis of both RO extracted and commercial alginates. This method is the most appropriate to determine the composition of the structure of alginate, obtaining the fractions of guluronic (G) and mannuronic (M) acid blocks that make up the alginate by comparison of the area of the spectrum signals corresponding to each M or G group, following Grasdalen et al. [36] Equations (6)–(9). A different peak is observed in the profile referring to the RO-extracted alginate at approximately 5.35 ppm, possibly due to impurities remaining from the algae during the alginate extraction process.
(6)$$F_{G}=\frac{I_{A}}{I_{B}+I_{C}}$$(7)$$F_{M}=1-F_{G}$$(8)$$M/G=\frac{1-F_{G}}{F_{G}}$$(9)$$F_{GG}=\frac{I_{C}}{I_{B}+I_{C}}$$
where *F_M_* is the fraction of *M* blocks in the alginate structure; *F_G_* is the fraction of *G* blocks; F_GG_ is the dyad sequence of *G* blocks; and *I_A_*, *I_B_,* and *I_C_* are the areas under the curve of the signals corresponding to G-1, (GM-5 + M-1) and G-5, respectively.

Thus, higher values of F_G_ and F_GG_ were obtained by the RO-extracted alginate (*F_G_* = 0.53 and *F_GG_
*= 0.53) compared to the commercial alginate (*F_G_* =0.47 and *F_GG_* = 0.23) and, therefore, lower values for the *M*/*G* in the RO extracted alginate (0.88) than in the commercial one (1.11). Similar results as the RO-extracted alginate were obtained by Larsen et al. (2003) [37] for alginate extracted from brown seaweeds *Sargassum latifolium* and *Sargassum asperiforium*. This structural difference implies a greater possibility of the formation of the structure known as the “egg box” produced by the cross-linking of Ca^2+^ ions with G-block fractions when calcium alginate gels are produced, thus improving its mechanical properties [38]. Also, authors such as Li et al. 2007, after studies carried out by X-ray diffraction, confirmed the presence of these structures, mainly when gelation occurs slowly in large pieces of gel, as is the case [39]. This structural difference between the alginate extracted from RO and the commercial one implies the possibility of obtaining materials with greater resistance using the same processing methods.

#### 3.1.2. Determination of Intrinsic Viscosity and Molecular Weight

Using Equation (1) [40], 12(ηr12−1) was plotted against $\frac{1}{C}$, observing in a linear dependence (R_2_ = 0.9994). Appendix A shows the correlation line between the viscosity and the concentration of the alginate solutions.

From the slope of the fitted straight line (0.4929), an intrinsic viscosity of 2.03 dL/g was obtained for the RO extracted alginate, and a value of 1.93 dL/g for the medium viscosity commercial alginate. Similar values were obtained for other brown seaweeds, such as *Laminaria digitata*, by Fertah et al. [18].

Once the intrinsic viscosity was known, the Mark Houwink Equation (2) was applied to determine the molecular mass of sodium alginate. A molecular mass of 94.874 kDa was obtained for the RO-extracted alginate and 90.241 kDa for the commercial medium viscosity alginate. These values are within the orders of magnitude obtained for other alginates extracted from seaweed and collected by Kaidi et al. [41]. These results, along with the ones obtained in Section 3.1.1, suggest that the sodium alginate extracted from the RO seaweed is suitable for some applications, as is the case of binding agents in tablets or thickening agents in gels and creams [42].

#### 3.1.3. DSC and TGA

Figure 3A shows the TGA profile of both the sodium alginate powder and the calcium alginate gel before and after the gelation process, respectively, as well as the first derivate for the sample extracted from RO. Figure 3B shows the DSC profile for these systems. This figure evidences the presence of four characteristic peaks in the first derivate signal when the samples are heated in an inert atmosphere. The first peak was observed at ~85 °C (it implies a loss of 10% of the initial weight), corresponding to the dehydration of the alginic salt.

Thus, comparing this result with the first peak observed in the TGA measurements, which corresponds to an endothermic event at ~75–95 °C, was previously attributed to hydrated salts [43].

Moreover, the DSC plot also shows an exothermic event at ~250 °C, which was also observed in the TGA profile (Figure 2A). This thermal event corresponded to the thermal decomposition of the biopolymer (found to be around 240–260 °C), which agrees with the results obtained by other authors such as Soares et al. [44] for commercial sodium alginate. The cross-linking of calcium with alginate on gels produced resistance to thermal degradation, which was reflected in a shift of this peak towards ~285 °C in TGA experiments when alginate is gelled (Figure 3A), and also a widening and delaying of the thermal event observed in the DSC profile (Figure 3B). This thermal decomposition produces the loss of about 20% of the initial mass of the sample of alginate and almost 30% in the case of the calcium alginate gel. From these values onwards, the mass loss responds to the formation of carbonate due to the decomposition of the carbonaceous materials. Similar results were observed by several authors [43,45] for calcium alginate nanoparticles used for drug delivery and biodegradable poly-(vinyl caprolactam) grafted onto alginate microgel, respectively.

### 3.2. Characterization of the Commercial and RO-Extracted Alginate Gels

#### 3.2.1. Rheological Characterization of Sodium Alginate Solution and Calcium Alginate Gels

Figure 4 displays the flow curves acquired for sodium alginate solutions at concentrations of 1% and 2% by weight (Figure 4A). Additionally, it illustrates the relationship between normalized viscoelastic moduli (G′/G′_0.1_ and G″/G′_0.1_, where G′ corresponds to the elastic modulus, G″ to the viscous modulus, and G′_0.1_ corresponds to the elastic modulus at 0.1 Hz) and frequency (Figure 4B). The figure also presents the variation in G′ and tan δ at 1 Hz (G′_1_, tan δ_1_, respectively) with the percentage of alginate at concentrations of 0.5%, 1%, and 2% by weight (Figure 4C). This analysis encompasses both commercial alginate variants (low and medium viscosity) and RO-extracted alginate gels. The flow curves obtained (Figure 4A) show an increase in viscosity values as the percentage of alginate in the solution increases. This result is common in other polymers and biopolymer solutions since the interaction between molecules increases when more molecules are in the solution [46,47]. Moreover, comparing the viscosity of the different alginates, the medium viscosity of commercial alginate is very similar to that of the extracted alginate, both being higher than the low viscosity commercial alginate (0.031 ± 0.0074, 3.8 ± 0.052, 2.8 ± 0.024 Pa·s for commercial low viscosity, medium viscosity, and RO-extracted alginate 2% at 1 s^−1^, respectively). The similarity of these values between commercial and RO-extracted alginate suggested that the number of G groups does not influence the rheology of sodium alginate solutions, as it does in calcium alginate gels due to the formation of “egg box” structure when the ions are replaced [29]. Moreover, this plot also evidenced that when there are limited interactions among biopolymer chains (curves exhibiting the lower viscosity) the solutions exhibited a Newtonian behaviour. However, when these interactions are greater, a shear thinning behaviour was observed since the values obtained decreased with the shear rate [48].

Figure 4B shows the G′ and G″ moduli normalized (i.e., G′/G′_0.1_ and G″/G′_0.1_) against frequency for the calcium alginate gels. According to the results obtained, all systems displayed an elastic behaviour as the storage modulus is always above the loss modulus, confirming the gelation of sodium alginate as calcium alginate following the procedure above-mentioned, regardless of the alginate nature analyzed, which agreed with the results obtained by LeRoux et al. [49] for calcium alginate gels. Furthermore, when comparing gels with the same percentage of polymer, the frequency dependence of commercial alginates was slightly higher than that of RO-extracted alginate. This distinction becomes clearer when examining the slope values for the normalized elastic modulus lines, these being 0.124 ± 0.005, 0.156 ± 0.003, and 0.110 ± 0.006 for gels prepared from commercial alginate (with low and medium viscosities) and RO-extracted alginate, respectively. This finding highlights a diminished frequency dependence of the viscoelastic moduli in gels prepared with RO in contrast to their commercial counterparts. Thus, the structure of the gel obtained from the RO-extracted alginate is expected to be less dependent on the relaxation time than the one formed by commercial alginate.

For a better understanding of the rheological behaviour of the calcium alginate gels obtained, the evolution of the elastic modulus (G′) and the tan δ at 1 Hz (G′_1_ and tan δ_1_, respectively) as a function of the alginate concentration (Figure 4C). First of all, when comparing the evolution of G′_1_ of the gels with the results of Figure 4A, it is observed for commercial medium viscosity alginate and RO-extracted alginate that, despite having identical viscosity values in solution, significant differences emerge in their elastic moduli when gelation occurs. Moreover, the gelation is confirmed by increase in viscosity, which can be estimated approximately by comparing the initial shear viscosity obtained solutions (e.g., around 0.36 and 3.91 Pa·s for the 1 and 2% extracted sodium alginate solutions, respectively) and the value of the complex viscosity (η*) obtained for those systems after gelation (i.e., around 19,800 and 58,300 Pa·s for the 1 and 2% extracted sodium alginate solutions, respectively).

Thus, this figure evidences that an increase in the percentage of alginate also involved an increase in the elastic modulus value (20,380, 32,400, and 57,400 Pa for commercial low viscosity, medium viscosity, and RO-extracted alginate 2% at 1 Hz, respectively). This figure confirms that the value of the elastic moduli was higher for gels made from RO-extracted alginate than those obtained for the medium-viscosity commercial alginate, which results in more marked solid behaviour for the RO-extracted calcium alginate gels. This result can be related to the higher number of “egg box” structures formed by the higher guluronic acid fractions in the structure of the extracted alginate chain compared to the commercial one, as previously reported in the ¹H-NMR analysis.

When the behaviour of tan δ_1_ is observed, a slight increase occurred when increasing the amount of alginate forming gels until, from 1% onwards, no significant differences are observed. This implies that there is no direct relationship between tan δ_1_ with greater chain cross-linking with increasing Ca^2+^ anchoring sites when the percentage of alginate in the gels is increased [49]. In summary, increasing the polymer concentration in the presence of Ca^2+^ produces an increase in G′ and stiffer gels with more solid behaviour, while at the same time, the tan δ_1_ value increases to a certain extent, due to a dissipation of intermolecular interactions and confirming an independence of the tan δ_1_ value with the Ca2+ concentration [49]. It is also interesting to note that the RO-extracted alginate gel has lower tan δ_1_ values than the commercial ones, which confirms a greater solid character than commercial calcium alginate gels at the same concentration. The results obtained by these sodium alginate gels agree with previous results reported by LeRoux et al. [49] for calcium alginate gels when they were analyzed by compressive and shear stresses. Thus, these results confirm the suitability of the alginate extracted from the invasive algae RO, which eventually led to higher mechanical properties than conventional alginates nowadays available for the industry.

#### 3.2.2. Water Uptake Capacity (WUC), Soluble Matter Loss (SML) and Water Holding Capacity (WHC)

Figure 5A shows the WUC and the SML of the three systems studied (RO-extracted alginate as well as low and medium viscosity commercial alginate) at different alginate concentrations (0.5, 1.0, and 2.0). It can be observed that WUC and SML values decreased when increasing the content of alginate in gels. When commercial alginates are analyzed, a significant decrease is observed with increasing viscosity. Nevertheless, although the extracted alginate has similar viscosity values to medium viscosity commercial alginate (Figure 4A), WUC values higher than these are observed, obtaining intermediate values between the commercial alginate of low and medium viscosity (1220 ± 163.3, 909.7 ± 21.6 and 1114.3 ± 106.4% for commercial low viscosity, medium viscosity and RO-extracted alginate gels at 1%, respectively). When the evolution of WUC was compared with the viscoelastic properties of the gels obtained (Figure 4C) (G′ in increasing order: low viscosity commercial, medium viscosity commercial and RO-extracted alginate), there was a general decrease in the WUC value as the amount of alginate in gels increased, but RO-extracted alginate gels display higher values of WUC than medium viscosity commercial alginate gels, even when G’ is higher for RO-extracted alginate gels. It is important to highlight that below 2% wt. alginate, hydrogels with superabsorbent capacity were obtained, with more than 1000% water uptake capacity [50]. Thus, these results confirm not only the increase in the rheological properties of the RO-extracted alginate but also the functional properties, as in the case of WUC.

SML values decrease as the alginate content in the gels increases (0.5% > 1% > 2%), as happened with WUC values. However, SML values are lower for RO-extracted alginate gels than for commercial ones, which implies more stable and less soluble structures than those. This is related to the higher G′ values (Figure 4C) and to the egg box structure that is formed in the case of RO-extracted alginate gels.

Figure 5B shows the evolution of WHC as a function of alginate content (0.5, 1 and 2 wt.%) for gels from both RO-extracted and commercial alginates. The same pattern is observed for the WUC values, decreasing as the percentage of alginate in the gels increases and showing higher values for RO-extracted alginate gels. These gels, despite having a higher G′ than the commercial alginate gel of medium viscosity, present WHC values more similar to that of low viscosity. Thus, the developed alginate structure had more free polar hydrophilic hydroxyl groups and carboxyl groups than the commercial medium viscosity one [51]. This structure developed not only allowed a suitable performance when mechanical properties were tested, but also these polar groups led to a high affinity for water, which determined the WUC and the WHC of the gels obtained.

#### 3.2.3. Scanning Electron Microscopy (SEM) of Calcium Alginate Gels

Figure 6 shows the SEM images obtained for the RO-extracted and commercial calcium alginate gels after freeze-drying. Figure 6A shows the gel obtained from the RO-extracted alginate, which was characterized by a slightly rough shape, with some pores in its structure. However, the surface commercial alginate observed in Figure 6B evidenced a far smoother surface with no visible pores on the surface. Alginate polymer chains contain hydrophilic groups (-OH or -COOH) in their structure, contributing to water absorption [52]. The porous structure in RO gel (Figure 6A) enhances water absorption due to increased surface area and favourable hydrophilic interactions, surpassing the commercial gel (Figure 6B). This greater porosity in Figure 6A than in Figure 6B can also explain the increase in water holding capacity in the RO gels, where significant differences were observed between the two gels (Figure 5B).

## 4. Conclusions

This study successfully extracted alginate from the invasive brown alga *Rugulopteryx okamurae* (RO), with a yield of approximately 74%. Characterization techniques employed confirmed the identity of the extracted alginate when compared with commercial brown seaweed alginate. The results obtained by RMN conducted to assume that the structure of the alginate obtained has a higher proportion of guluronic acid (G) than mannuronic acid (M) (0.53 and 0.47 for extracted and commercial medium viscosity alginate, respectively), which favoured the formation of “egg box” structures by replacing sodium with calcium during gelation. The mechanical spectra obtained showed a prevailing elastic behaviour in the frequency range studied for all the samples analyzed. By increasing the weight percentage of alginate in each gel, an increase in the elastic modulus was observed. Thus, elastic modulus at 1 Hz increased from 3.69 ± 0.56 to 58.20 ± 1.12 kPa for RO-extracted alginate, from 1.91 ± 0.15 to 32.40 ± 2.90 to for medium viscosity commercial alginate, and from 1.12 ± 0.013 to 17.37 ± 4.26 kPa for low viscosity commercial alginate when alginate increased from 0.5 to 2%, respectively. It also showed a higher frequency dependence with increasing concentration, except for the RO-extracted alginate gel, which showed an even lower frequency dependence. These results were consistent with the tan δ values, which increased with alginate content in the gels, obtaining lower tan δ values at 1 Hz for the RO-extracted alginate (0.17 ± 0.0056, 0.23 ± 0.011 and 0.16 ± 0.0035 for the reference low viscosity and medium viscosity commercial alginate and RO-extracted commercial alginate at 1%, respectively). These results indicate that the gelation of the extracted alginate produces gels that are stiffer and have a more solid-like character than the two commercial alginates used as references in this work. With increasing elastic modulus and alginate content, there was a general decrease in water absorption and retention values. The extracted alginate showed higher water absorption values compared to the medium viscosity commercial alginate, with values of 1114.3 ± 106.4% at 1% RO-extracted alginate, suggesting potential for obtaining superabsorbent gels. While preliminary findings suggest superior structural and mechanical properties of alginate extracted from Rugulopteryx okamurae compared to two commercially available alginates, it is essential to acknowledge the need for further extensive research to substantiate these claims. Initial results provide promising insights, warranting continued investigation into the potentially enhanced qualities of *Rugulopteryx okamurae*-derived alginate for various applications.

## Figures and Tables

**Figure 1 polymers-16-00702-f001:**
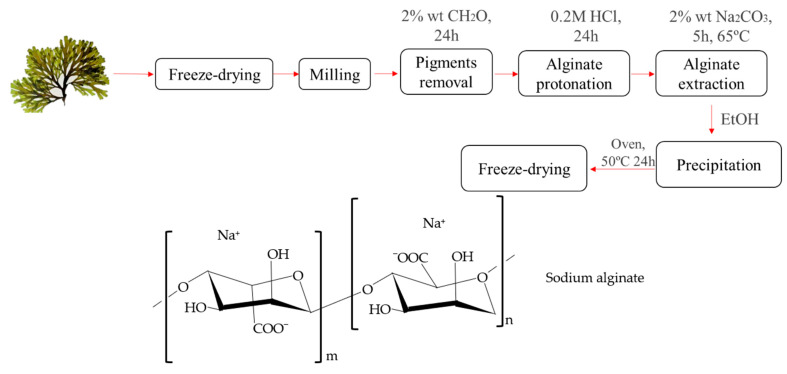
Flow chart of the sodium alginate extraction from *Rugulopteryx okamurae* and structure of sodium alginate.

**Figure 2 polymers-16-00702-f002:**
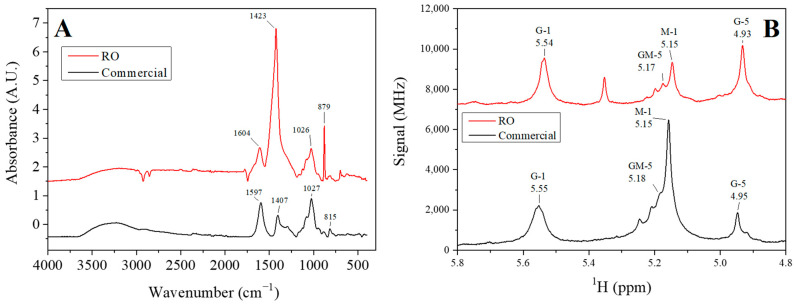
Fourier Transform Infrared analysis (**A**) and ¹H-NMR analysis (**B**) of commercial and *Rugulopteryx okamurae* (RO) extracted alginate.

**Figure 3 polymers-16-00702-f003:**
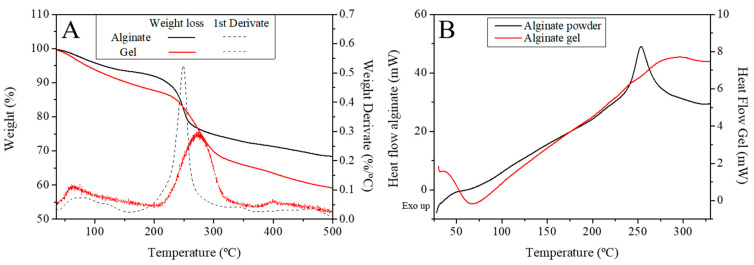
Thermogravimetric Analysis (TGA) (**A**) and Differential Scanning Calorimetry (DSC) (**B**) of the *Rugulopteryx okamurae* (RO) extracted alginate made under N_2_ atmosphere.

**Figure 4 polymers-16-00702-f004:**
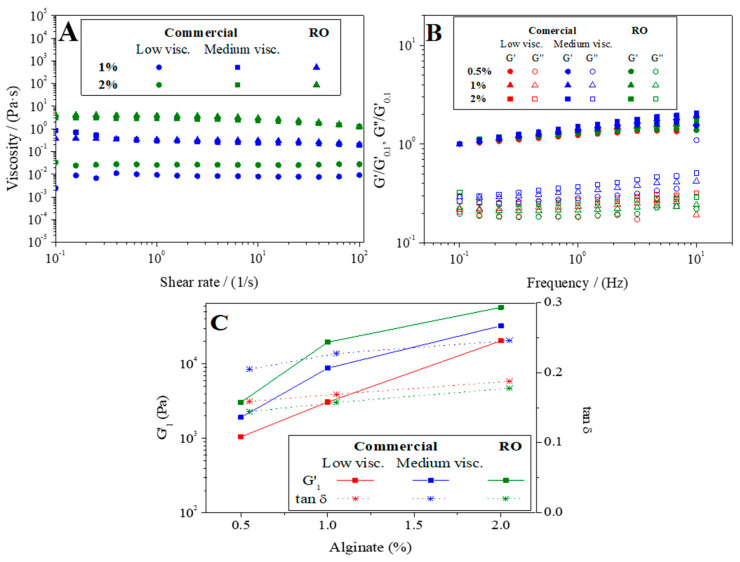
(**A**) Evolution of viscosity with share rate for different systems before gelation prepared at 1 and 2% wt.; (**B**) evolution of G′/G′_1_ and G″/G′_1_ modulus with the frequency of commercial low and medium viscosity alginate gels and extracted alginate gel from *Rugulopteryx Okamurae* (RO), prepared at 0.5, 1, and 2% wt; and (**C**) evolution of G′_1_ and tan d with the percentage of alginate forming gels.

**Figure 5 polymers-16-00702-f005:**
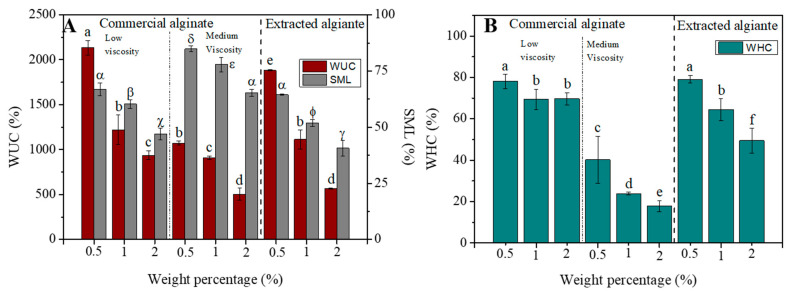
Water uptake capacity (WUC) and soluble matter loss (SML) (**A**), and water holding capacity (WHC) (**B**) of commercial low and medium viscosity alginate gels and extracted alginate gel from *Rugulopteryx Okamurae* (RO). Different letters within a column indicate significant differences (*p* < 0.05).

**Figure 6 polymers-16-00702-f006:**
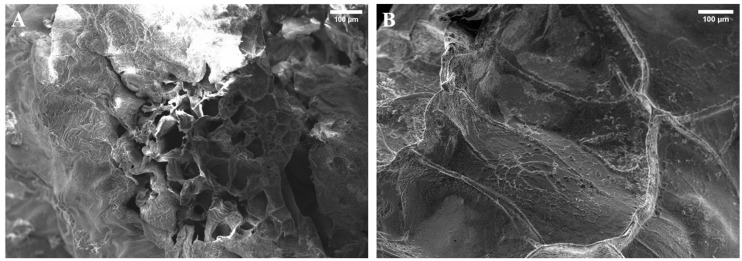
SEM images of calcium alginate gels from alginate extracted from *Rugulopteryx Okamurae* (RO) (**A**) and medium viscosity commercial alginate (**B**) after freeze-drying at 100× magnification.

## Data Availability

Data are contained within the article.

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
