# Peer review of "Feasibility of Invasive Brown Seaweed Rugulopteryx okamurae as Source of Alginate: Characterization of Products and Evaluation of Derived Gels"

_polymers, 2024, doi:10.3390/polym16050702_

Round 1

Reviewer 1 Report

Comments and Suggestions for Authors

The manuscript entitled "Feasibility of Invasive Brown Seaweed Rugulopteryx okamurae as Source of Alginate: Assessment of Its Gelling Properties" delas with the extraction of alginate from an invasive seaweed, and the characterization of its gelling behavior.

in mt view, the topic of the submitted work is quite interesting, and the followed experimental procedure is well-designed.

However, before recommending the manuscript for publication, the following issues need to be adressed:

- Please, enhance the quality of figures 3 and 4, as the legend is hardly readable.

- In the comment about TGA results, the Authors stated that the cross-linking of alginate improved its thermal oxidation resistance. However, in the experimental part is reported that TGA was performed in nitrogen N2, therefore only thermal degradation can be evaluated through such kind of test. Please correct the comment.

-  Please explain the normalization performed to obtain normalized viscoelastic moduli.

- Page 8, lines 300-304: the sentence is quite confusing, please rephrase.

- The Authors stated that the frequency dependence of the commercial alginates is hogher than that of RO-extracted one; however, this feature is not clearly observable in Figure 4b. Please, provide evidences for this behavior. 

- Please, check the meaning of the sentence at page 8, lines 330-332.

- In the comment about Figure 4C, the Authors refers to viscosity, but only G' and tan_delta are plotted in the Figure.

Comments on the Quality of English Language

Please, check the whole manuscript in order to correct all typos and grammar errors.

Reviewer 2 Report

Comments and Suggestions for Authors

 1. Publication sections 2.2.5 and 3.1.2 are largely identical. The descriptive part of the experiment from the “discussion of results” chapter, section 3.1.2 should be moved to the experimental part, section 2.2.5. In section 3.2.1, leave only a discussion of the results obtained.

2. Section 3.1.2 indicates “supplementary Figure 1”. This figure is not included in the text of the publication.

3. In section 2.4.3. there is the phrase “drying samples in an oven at 50ºC overnight.” Not specific at all!!! Either time should be specified or, most often, dried to constant weight.

4. SEM photos were taken at low magnification, so pores in the samples are not visible. If you zoom in higher, the pores will be visible. The pore size can be only 1-10 microns. In this regard, the authors’ comments “water absorption is mainly carried out by the hydrophilic groups -OH and -COOH of the polymer chains, and not by the pores”, in my opinion, is not correct.

5. Conclusions based on research results are not structured and are too lengthy. The conclusions should be shortened by indicating the results in accordance with the purpose and objectives of the study.

6. In the list of references, more than half of the publications are on studies performed before 2018. The reference list should be updated with new publications on the research topic. This is especially true for links in "Introduction". The work belongs to a very relevant area of research!

Reviewer 3 Report

Comments and Suggestions for Authors

GENERAL COMMENTS

The work is of considerable interest for the exploitation of natural resources and should, therefore, reported in the literature subject to substantial revisions. The interpretation of results is often too verbose and consisting of tedious descriptions of information from graphs, rather than focussing on the salient features of the data.  The presentation consists mostly of factual descriptions of results and the terminology is sometime inaccurate. There are also several exaggerated claims that not only could be misleading the reader but would also downgrade the value of the work. A major shortcoming is the lack of clarity about the originality of the procedure used for the treatment of the algae. Furthermore, Authors have not attempted to explain the gelation mechanism and consider the possible role of the ratio of the cations Na+ / Ca 2+ as a possible controlling parameter of gelation.  This would be expected from a competition of the two cations for the same available quantity of anions in the polymer chains.

COMMENTS ON SPECIFIC SECTIONS

TITLE: I suggest changing the wording in the subtitle to reflect more accurately the work presented may be something like ….” Characterization of products and evaluation of derived gels”

KEYWORDS: Gel and Rheology are too general.

ABSTRACT: (a) It is difficult to identify the claimed 74 % yield in the data presented in the text. (b)There are statements on structural aspects that are not strongly supported from the data obtained and there are too many numerically specific data, but they are not misleading. (c) The claims in lines 21 – 24 are neither necessary nor are substantiated by the data obtained from the work.  There would have to be a lot of more work to be done to attract the attention of those likely to have an interest in the procedure used.

INTRODUCTION:  In the paragraph at the end there are statements on the work done but not on specific objectives and on the novelty aspects. There should be a clear elucidation of how the work undertaken differs from that already reported and the about the rationale for the procedure used.

MATERIALS AND METHODS:  Section 2.1 - 1) State the origins of the procedure outlined in Figure 1 and clarify whether it is an adaptation of some previously used procedures or whether it is entirely new. Authors should explain how the treatment with CH2O is expected to “remove” the pigments. Are chemical reactions expected to take place with specific groups in RO, which would destroy the chromophore groups in the pigment.  2) It would be useful to show the structure of the sodium alginate and the gelation mechanism for the gelation caused by the introduction of Ca2+ ions. Section 2.2 – State the source of the commercial sodium alginate with an indication of its molecular weight (molar mass) and % purity. Sections 2.4.2, 2.4.3 and 2.4.4 - Describe a) how the cylindrical specimens (25 mm high and 20 mm diameter) were made, b) how they were cut into this discs (presumably) for the rheological measurements, and c) what was the water content of the gels, which would have a large effect on the results.

RESULTS AND DISCUSSION: Section 3.1.1 – The traces in both A and B diagrams in Figure 2 show that there are distinct discrepancies between the RO extracted products and the commercial Na alginate that have not been discussed. For instance, A) While most of the peaks in the H-NMR traces occur at the same H(pm) for both systems, the one at ~ 5.35 for the RO sample does not feature in the commercial Na alginate. This must be due to the presence of a significant amount of a different component, B) Regarding the ATR spectra only the peaks at 1026 and, possibly, at 1600 cm -1 can be perceived to be common to both samples. At the same time, the ratio of the hights of these two internal peaks (same sample) are approximately equal for the two materials as a confirmation of the presence of sodium alginate in both systems. But the others are so different and can only be attributed to the presence of large quantities of different species present in the RO sample.            Section 3.1.2  - (a) The values for viscosity should also be given in SI units. (b) The values for the molar mass (molecular weight) are probably underestimated by a factor of 1000 and the units should be in Dalton, even though the numerical value is the same as in g/mol. (c) In lines 265 – 269 state what are the specific characteristics of the extracted products that would make them suitable for use as bonding agents in gels and creams. (d) The statement in lines 267 -269 does not say anything new or useful. Section 3.1.3 -  (a) It is not clear what is the difference between “powder” and “gel” and the legend is too small to read it. (b) There is no peak at 85-95 C in neither DSC traces (presumably associated with the release of water from the samples). (c) It is not clear what the subscript 1 for G’ and G” refer to. Presumably normalized to 0.1 Hz, i.e. G’(f)/G’(0.1). Why not writing it this format to make it clear? (Line 320, should be “moduli” and not modules. (d) In lines 333 – 335 the viscosity values are likely to be for solutions and not gels. The viscosity of irreversible gels is “infinite”, according to the definition of gel, unless the network is broken by the applied stress to induce flow. Please clarify. (e) There is no need for Fig 4B as there are no new insights emerging from the data, which are difficult to follow in any case. (f) It is difficult to understand the meaning of G’1 (and presumably also tanδ1) in Fig 4C. If these refer to normalised values, i.e. G’(f)/G’(0.1), then they cannot be constant for a given system because of the frequency dependence. (g) In lines 362 – 365 elaborate the discussion relative to the comparison of your data with those reported in ref 45. (k) The statement in lines 364 – 365 is not appropriate as there is no evidence in your data for this extraordinary claim.

CONCLUSIONS: (a) The comment about the 74 % yield (not assay) has already been made for the Abstract. (b) The claims in lines 427 – 430 are not strongly supported by the data presented in the text. (c) The statements in lines 444 – 446 are too speculative to be reported in the Conclusions. (d) The last sentence (lines 454 – 457) cannot be justified because (i) the comparison was made using only one commercial grade of Na alginate, and (ii) there are insufficient data in the text to make the stated generalization.

Comments on the Quality of English Language

Mostly OK but the terminology is sometime inaccurate

Round 2

Reviewer 1 Report

Comments and Suggestions for Authors

I recommend the publication of the manuscript as it stands, since it was significantly modified following the Reviewers' suggestions.

Reviewer 3 Report

Comments and Suggestions for Authors

All comments have been addressed and suggestions have been implemented. I have not seen the structure of the gelled products in the graphical abstract, but I am sure it’s fine.